A novel long non-coding RNA, AC012456.4, as a valuable and independent prognostic biomarker of survival in oral squamous cell carcinoma

Hu Xuegang 1 2
Qiu Zailing 1 2
Zeng Jianchai 1 2
Xiao Tingting 1 2
Ke Zhihong 1 2
Lyu Hongbing hongbinglu@fjmu.edu.cn hongbinglu@126.com 1
1 Department of Endodontics and Operative Dentistry, School and Hospital of Stomatology, Fujian Medical University , Fuzhou , China
2 Key laboratory of Stomatology, Fujian Province University , Fuzhou , China
Uversky Vladimir
Electronic publication date: 2018 Aug 13
Publication date: 2018
Volume: 6
Electronic Location ID: e5307
Received 2018 Mar 8; Accepted 2018 Jul 4
Copyright: ©2018 Hu et al.
Copyright year: 2018
Copyright holder: Hu et al.
License: This is an open access article distributed under the terms of the Creative Commons Attribution License, which permits unrestricted use, distribution, reproduction and adaptation in any medium and for any purpose provided that it is properly attributed. For attribution, the original author(s), title, publication source (PeerJ) and either DOI or URL of the article must be cited.
License URL: https://creativecommons.org/licenses/by/4.0/

Keywords: Long non-coding RNAs (lncRNAs), Oral squamous cell carcinoma (OSCC), AC012456.4, Prognostic biomarkers

Funding: The authors received no funding for this work.

==============================
Oral squamous cell carcinoma (OSCC) is a major malignant cancer of the head and neck. Long non-coding RNAs (lncRNAs) have emerged as critical regulators during the development and progression of cancers. This study aimed to identify a lncRNA-related signature with prognostic value for evaluating survival outcomes and to explore the underlying molecular mechanisms of OSCC. Associations between overall survival (OS), disease-free survival (DFS) and candidate lncRNAs were evaluated by Kaplan–Meier survival analysis and univariate and multivariate Cox proportional hazards regression analyses. The robustness of the prognostic significance was shown via the Gene Expression Omnibus (GEO) database. A total of 2,493 lncRNAs were differentially expressed between OSCC and control samples (fold change >2, p < 0.05). We used Kaplan–Meier survival analysis to identify 21 lncRNAs for which the expression levels were associated with OS and DFS of OSCC patients (p < 0.05) and found that down-expression of lncRNA AC012456.4 especially contributed to poor DFS (p = 0.00828) and OS (p = 0.00987). Furthermore, decreased expression of AC012456.4 was identified as an independent prognostic risk factor through multivariate Cox proportional hazards regression analyses (DFS: p = 0.004, hazard ratio (HR) = 0.600, 95% confidence interval(CI) [0.423–0.851]; OS: p = 0.002, HR = 0.672, 95% CI [0.523–0.863). Gene Set Enrichment Analysis (GSEA) indicated that lncRNA AC012456.4 were significantly enriched in critical biological functions and pathways and was correlated with tumorigenesis, such as regulation of cell activation, and the JAK-STAT and MAPK signal pathway. Overall, these findings were the first to evidence that AC012456.4 may be an important novel molecular target with great clinical value as a diagnostic, therapeutic and prognostic biomarker for OSCC patients.

Introduction

The five-year survival rate is approximately 50% for oral squamous cell carcinoma (OSCC), which is one of the most common malignancies of the head and neck region (Bozec et al., 2009; Ferlay et al., 2015; Kamangar, Dores & Anderson, 2006; Kim et al., 2017; Verusingam et al., 2017). The predisposition of OSCC to distant metastases and metastases in the lymph nodes, its highly invasive nature, and its tendency towards local recurrence are important factors that contribute to the poor prognosis of OSCC patients (Massano et al., 2006; Singh & Schenberg, 2013). Hence, more effective novel tumor diagnostic and prognostic biomarkers (Mehrotra & Gupta, 2011), which can improve the survival rate and can be used to assess treatment outcomes, are urgently needed.

The Cancer Genome Atlas (TCGA) (http://cancergenome.nih.gov) database, which is primarily used to collate specimens from cancer patients and adjacent normal tissue specimens, contains large data sets collected with high-throughput methods at multiple genomic and proteomic levels (Chin, Andersen & Futreal, 2011; Wang, Gerstein & Snyder, 2009). The Gene Expression Omnibus (GEO, http://www.ncbi.nlm.nih.gov/geo/) is the largest and most comprehensive public gene expression repository for high-throughput data at NCBI (Barrett & Edgar, 2006; Clough & Barrett, 2016). Both the GEO and TCGA collect macroscopic clinical information, such as stage and grade of tumor, survival time, age, sex, and race. Therefore, the TCGA and GEO databases can be analyzed systematically and comprehensively to explore important potential value and information.

In this study, we first sought to use the existing GEO microarrays and TCGA RNA-seq data to identify differential expression of lncRNAs between OSCC and control tissue samples. Then, the differentially expressed lncRNAs were evaluated by Kaplan–Meier survival analysis and univariate, multivariate Cox proportional hazards regression analyses and Gene Set Enrichment Analysis (GSEA). Ultimately, through systematic and objective analysis, we first discovered that lncRNA AC012456.4 is significantly associated with survival outcomes of OSCC patients based on TCGA data. Then, AC012456.4 was further successfully confirmed as a potential prognostic biomarker for the prediction of overall survival (OS) in the GEO database. We hope that the lncRNA AC012456.4 revealed in our study may serve as a novel biomarker and potential target for the diagnosis, treatment, and prognosis of OSCC.

Materials and Methods

Data source

The RNA-seq data and corresponding patient information data of head and neck cell carcinoma (HNSC) were downloaded from the TCGA database. Clinical samples from the oral cavity (buccal mucosa, tongue, lip, hard palate, alveolar ridge, floor of the mouth and oral cavity) were chosen, while some samples from other parts (hypopharynx, larynx, oropharynx and tonsil, for example) were excluded. The original microarray data between OSCC and adjacent normal tissue samples were downloaded from the NCBI GEO databases. The accession numbers were GSE36820 and GSE41613, respectively. The microarray data of GSE36820 and GSE41613 were based on GPL570 (Affymetrix Human Genome U133 Plus 2.0 Array).

Data pre-processing and differential expression analysis

The edgeR package was downloaded from the Stanford University website. The original microarray data from the GEO were converted into expression measures using the affy R package. Then, the differentially expressed lncRNAs were identified by the Limma R package (Ritchie et al., 2015; Teufel et al., 2016). The differentially expressed lncRNAs that were screened from the TCGA were analyzed by the edgeR package (Robinson, McCarthy & Smyth, 2010). To improve screen accuracy and simplify the screening process, the cut-off criteria, which was in accordance with the procedure of Benjamini & Hochberg (BH), was as follows: 1. the false discovery rate was controlled at 0.01; 2. the fold change should be more than 2. The differentially expressed lncRNAs among GSE36820, GSE41613 and the TCGA were identified by the intersect function in the R package. Tumor and normal tissue data were recorded and were statistically analyzed.

Identification of lncRNAs with prognostic value in OSCC

The differences between expressed lncRNAs (fold change >2, p < 0.05) are involved in the prognostic value for OSCC. The OSCC patients were divided into two parts, depending on the average expression level of candidate lncRNAs: a high expression group and a low expression group. Survival differences and p-values were compared between the two groups and were evaluated using a Kaplan–Meier survival analysis and a log-rank test. After this, a univariate Cox proportional hazards regression analysis (Bair & Tibshirani, 2004) was conducted to assess the correlation between candidate lncRNAs and patient overall survival (OS) and disease-free survival (DFS) (p < 0.05). Statistically significant lncRNAs and clinical candidate predictors were further evaluated by multivariate Cox proportional hazards regression analyses to identify independent prognostic lncRNAs. Candidate predictors included age, gender, grade, and stage. We then performed subgroup analyses. The hazard ratio (HR) and 95% confidence interval (CI) were also assessed.

Gene set enrichment analysis (GSEA)

GSEA 2-2.2.3 (JAVA version) was downloaded from the Gene Set Enrichment Analysis website (http://software.broadinstitute.org/gsea/index.jsp). Then, the downloaded dataset was imported using the GSEA software. Gene sets identified as related to biological signal conduction on the MSigDB (Molecular Signatures Database) (http://software.broadinstitute.org/gsea/msigdb), which may be found on the GSEA website, served as reference gene sets. This process was repeated 1,000 times for each analysis according to the default weighted enrichment statistical method. Gene sets with a false discovery rate (FDR) <0.25 and a family-wise error rate (FWR) <0.05. The GSEA analysis includes four key statistics: Enrichment Score (ES), Normalized Enrichment Score (NES), False Discovery Rate (FDR) and P-value.

Statistical analysis

In this study, all analyses, including the t-test, heat map, and survival analyses, were performed with the R, GraphPad and SPSS software packages. p values less than 0.05 were considered significant. All statistical tests were two-sided.

Results

Characteristics of OSCC patients according to the TCGA

In this study, the datasets of 350 OSCC patient and 44 controls were acquired and downloaded from the TCGA (http://cancergenome.nih.gov) database; these datasets contained expression data and clinical information related to 14,448 lncRNAs. The clinicopathological features of all patients are shown in Table 1. The mean ±  standard deviation (STDEV) for all patient ages is 61.590 ±  12.886.

Table 1 The clinicopathological characteristics of patients from the TCGA database.

Characteristics	Number of case	No. of patients (%)	
Age (years)	346		
≦60		152(41.33%)	
≧60		194(58.67%)	
Median (range)		61.590(19–90)	
Gender	347		
Male		236(68.01%)	
Female		111(31.99%)	
Alcohol history	339		
No		111(32.74%)	
Yes		228(67.26%)	
Perineural invasion present	263		
No		123(46.77%)	
Yes		140(53.23%)	
Margin status	324		
Close		39(12.04%)	
Negative		244(75.31%)	
Positive		41(12.65%)	
Lymphovascular invasion present	250		
Yes		76(30.40%)	
No		174(69.60%)	
Tumor stage	314		
Stage I		21(6.69%)	
Stage II		56(17.83%)	
Stage III		64(20.38%)	
Stage IV		173(55.10%)	
T stage	335		
T1		34(10.15%)	
T2		103(%)	
T3		70(%)	
T4		128(%)	
N stage	334		
NO		126(37.72%)	
N1		52(15.57%)	
N2		110(32.93%)	
N3		46(13.77%)	
M stage	170		
M0		125(73.53%)	
M1		45(26.47%)	
Histologic grade	344		
G1		53(15.41%)	
G2		210(61.05%)	
G3		71(20.64%)	
G4		10(2.91%)	
Vital status	347		
Alive		227(65.42%)	
Dead		120(34.58%)	

Significant differentially expressed lncRNAs in OSCC

In all, 2,493 differentially expressed lncRNAs were identified through analysis of 14,448 lncRNAs using the edgeR packages (fold change >2, p < 0.05) (Fig. 1). Moreover, 855 lncRNAs were down-regulated and 1,638 lncRNAs were up-regulated in the OSCC samples compared to normal tissue. Down-regulated and up-regulated lncRNAs account for 34.2% and 65.6% of the differentially expressed lncRNAs, respectively.

Figure 1 A heat map drawn to show differential lncRNA expression in OSCC and normal tissue samples from the TCGA datasets, which were analyzed with R software.

Representative genes of each cluster were selected and represented as a heat map. Genes shown in red are upregulated and genes in blue are downregulated. The magnitude of the regulation is illustrated by the intensity of the color.

Identification of survival differences lncRNAs in OSCC

We used a Kaplan–Meier survival analysis with the log-rank test to identify relationships between the above 2,493 lncRNA signatures and the survival of OSCC patients. Then, we determined the levels of 21 lncRNA signatures that were significantly related to OS and DFS. Among these 21 lncRNAs, a significant positive correlation was observed between the signatures of 13 lncRNAs (TTC39A-AS1, RP11-93B14.9, AC012456.4, RP11-87C12.5, RP11-464F9.21, LINC01549, RP11-897M7.1, AP003900.6, LINC01343, RP11-181E10.3, CTD-2545H1.2, RP11-796E2.4 and LINC01108) and OS/DFS. In contrast, the signatures of the remaining 8 lncRNAs (AC007879.2, BOK-AS1, CTB-161M19.4, CTD-2033A16.3, FAM95B1, RP11-1C8.7, RP11-285G1.14 and RP11-286E11.1) were significantly negatively correlated with OS and DFS. That is, low expression of the 13 lncRNAs described above correlated with a poor prognosis of OSCC patients, while the up-regulation of the latter 8 lncRNAs correlated with a shorter survival time (Fig. 2) (Table 2).

Figure 2 Kaplan–Meier survival analyses and log-rank tests for OS and DFS in OSCC.

(A) OS and (B) DFS rates of all patients according to AC012456.4 expression. (C) OS and (D) DFS rates of all patients according to AP003900.6 expression. (E) OS and (F) DFS rates of all patients according to BOK-AS1 expression. (G) OS and (H) DFS rates of all patients according to LINC01108 expression. (I) OS and (J) DFS rates of all patients according to RP11-1C8.7 expression. (K) OS and (L) DFS rates of all patients according to RP11-87C12.5 expression.

Table 2 Twenty-one lncRNA levels significantly correlated to OS and DFS.

LncRNA	Gene ID	Chromosome	OS (P value )	DFS (P value )	
AC012456.4	ENSG00000230790	chr2	0.00987	0.00828	
AP003900.6	ENSG00000271308	chr21	0.00868	0.00397	
BOK-AS1	ENSG00000234235	chr2	0.01812	0.01597	
LINC01108	ENSG00000226673	chr6	0.00631	0.00767	
RP11-1C8.7	ENSG00000271830	chr8	0.00035	0.04009	
RP11-87C12.5	ENSG00000255856	chr12	0.01058	0.00048	
TTC39A-AS1	ENSG00000261664	chr1	0.04276	0.00371	
RP11-93B14.9	ENSG00000277496	chr20	0.01279	0.00352	
AC007879.2	ENSG00000234902	chr2	0.00811	0.03607	
RP11-464F9.21	ENSG00000234606	chr10	0.01486	0.03221	
LINC01549	LINC01549	chr21	0.00021	0.0165	
CTB-161M19.4	ENSG00000249494	chr5	0.04807	0.01152	
RP11-286E11.1	ENSG00000245293	chr4	0.03618	0.0041	
RP11-897M7.1	ENSG00000256209	chr12	0.03129	0.02265	
LINC01343	ENSG00000237290	chr1	0.01115	0.03191	
FAM95B1	ENSG00000223839	chr9	0.04778	0.01648	
RP11-181E10.3	ENSG00000271590	chr2	0.00597	0.00934	
CTD-2545H1.2	ENSG00000262445	chr17	0.02892	0.02929	
RP11-796E2.4	ENSG00000245904	chr12	0.04276	0.00371	
CTD-2033A16.3	ENSG00000262136	chr16	0.04586	0.02714	
RP11-285G1.14	ENSG00000273363	chr10	0.01276	0.00503	

Through the above Kaplan–Meier survival analysis, the variables of age, gender, grade, tumor stage, and TNM stage were identified as statistically significant factors that are related to the above 21 lncRNAs and patient prognosis. We also applied univariate and multivariate Cox regression analyses to evaluate the ability of 21 candidate lncRNA signatures to serve as independent prognostic variables. The univariate analysis indicated that decreased AC012456.4 expression (HR = 0.706, 95% CI [0.551–0.903], p = 0.006), age, tumor stage, and TNM stage were all significantly related to worse OS in OSCC patients (Table 3). Decreased AC012456.4 expression (HR = 0.601, 95% CI [0.423–0.853], p = 0.004) was the only variable that could predict poorer DFS for OSCC. Finally, multivariate Cox regression analysis revealed that low expression of AC012456.4 was the only independent prognostic variable for both OS (HR = 0.672, 95% CI [0.523–0.863], p = 0.002) and DFS (HR = 0.600, 95% CI [0.423–0.851], p = 0.004) in OSCC patients (Table 4). In addition, age and N stage were highly significantly correlated with shorter OS or DFS.

Table 3 Univariate and multivariate Cox regression analysis for OS in patients with OSCC.

Variables	Univariate analysis	Multivariate analysis	
	P value	HR	95% CI	P value	HR	95% CI	
Age (years)	0.003	1.021	1.007, 1.036	0.001	1.026	1.011, 1.041	
Gender	0.459	1.150	0.794, 1.665	0.481	1.145	0.786, 1.666	
Grade	0.127	1.215	0.946, 1.560	0.062	1.276	0.988, 1.648	
Stage	
(age ≦ 60)	0.034	1.425	1.026, 1.978	0.210	0.765	0.503, 1.163	
(age >  60)	0.523	1.080	0.853, 1.367				
N	0.015	1.263	1.046, 1.524	0.011	1.279	1.059, 1.546	
T (age ≦ 60)	0.003	1.551	1.160, 2.075	0.293	1.101	0.921, 1.316	
  (age > 60)	0.873	0.982	0.783, 1.230				
AC012456.4	0.006	0.706	0.551, 0.903	0.002	0.672	0.523, 0.863	
Notes.

N Regional Lymph Nodes

T Primary Tumor

Table 4 Univariate and multivariate Cox regression analysis for DFS in patients with OSCC.

Variables	Univariate analysis	Multivariate analysis	
	P value	HR	95% CI	P value	HR	95% CI	
Age (years)	0.093	1.017	0.997, 1.036	0.071	1.018	0.999, 1.037	
Gender	0.627	1.132	0.687, 1.867	0.678	1.113	0.672, 1.841	
Grade	0.817	1.043	0.732, 1.485	0.533	1.125	0.777, 1.627	
Stage	0.625	1.064	0.830, 1.363	0.482	0.852	0.545, 1.332	
N	0.539	1.085	0.7837, 1.407	0.167	1.286	0.900, 1.836	
T	0.191	1.167	0.926, 1.470	0.295	1.134	0.896, 1.434	
AC012456.4	0.004	0.601	0.423, 0.853	0.004	0.600	0.423, 0.851	

lncRNA AC012456.4 was low expressed in OSCC tissues and associated with clinicopathological parameters

OSCC patients were further classified into high or low expression groups based on the median value of the relative lncRNA expression. The expression of lncRNA AC012456.4 was significantly weaker in OSCC tissue samples (1.360 ± 0.05569) relative to normal tissue samples (3.062 ± 0.2304) in the TCGA (p < 0.0001) (Fig. 3). The correlation between lncRNA AC012456.4 expression and clinicopathologic parameters of OSCC patients was also further analyzed. As shown in Table 5, lncRNA AC012456.4 expression was significantly correlated with alcohol history consumption (p = 0.033). Additionally, decreased expression of lncRNA AC012456.4 expression was nearly significantly associated with T stage (p = 0.075). However, no significant association was found between other clinicopathological factors and lncRNA AC012456.4 expression.

Figure 3 Expression of AC012456.4 in normal tissues and OSCC tissues.

AC012456.4 expression is significantly down-regulated in OSCC samples (1.360 ± 0.05569) in comparison to adjacent non-cancerous tissues (3.062 ± 0.2304) in the TCGA dataset.

Table 5 AC012456.4 expression and clinicopathological characteristics of patients with OSCC.

Characteristics	Number of case	AC012456.4 expression	P value	
		Decreased number (%)	Non-decreased number (%)		
Age (years)				0.082	
≥60	186	96(51.61%)	90(48.39%)		
<60	143	60(41.96%)	83(58.08%)		
Gender				0.745	
Female	102	47(46.08%)	55(59.92%)		
Male	227	109(48.02%)	118(51.98%)		
Alcohol history				0.033	
Yes	213	109(51.17%)	104(48.83%)		
No	104	40(38.46%)	64(61.54%)		
M stage				0,511	
M0	119	56(47.06%)	63(52.94%)		
M1	39	16(41.03%)	23(58.97%)		
T stage				0.075	
T1  + T2	128	54(42.19%)	74(57.81%)		
T3  + T4	189	99(52.38%)	90(47.62%)		
N stage				0.163	
N0  + N1	168	87(51.79%)	81(48.21%)		
N2  + N3	148	65(43.92%)	83(56.08%)		
Notes.

M0 No distant metastasis (no pathologic M0; use clinical M to complete stage group)

M1 Distant metastasis

N0 No regional lymph node metastasis

N1 Metastasis in a single ipsilateral lymph node, 3 cm or less in greatest dimension

N2 Metastasis in a single ipsilateral lymph node, more than 3 cm but not more than 6 cm in greatest dimension; or in multiple ipsilateral lymph nodes, none more than 6 cm in greatest dimension; or in bilateral or contralateral lymph nodes, none more than 6 cm in greatest dimension

N3 Metastasis in a lymph node more than 6 cm in greatest dimension

T1 Tumor 2 cm or less in greatest dimension

T2 Tumor more than 2 cm but not more than 4 cm in greatest dimension

T3 Tumor more than 4 cm in greatest dimension

T4a Moderately advanced local disease

T4b T4b Very advanced local disease

Tumor invades masticator space, pterygoid plates, or skull base and/or encases internal carotid artery.

Evaluation of the prognostic value of lncRNA AC012456.4 via the GEO

For the purpose of evaluating the robustness of lncRNA AC012456.4 expression in the prediction of OS of OSCC patients, we acquired other independent datasets from the GEO with accession numbers of GSE36820 and GSE41613, which contained OSCC samples, but samples with incomplete clinical information were excluded. The prognostic signatures and the Kaplan–Meier analysis were calculated and performed for each OSCC sample. In agreement with the result of the TCGA datasets, low expression levels of lncRNA AC012456.4 were associated with lower OS (Fig. 4). The lncRNA AC012456.4 was also expressed at low levels in OSCC tissues (p < 0.0001).

Figure 4 Evaluation of the prognostic value of lncRNA AC012456.4 via the GEO.

(A) Heatmap of lncRNA AC012456.4 expression in GEO. (B) lncRNA AC012456.4 expression was significantly low in OSCC. (C) OSCC patients were divided into the high expression group and the low expression group according to the median lncRNA AC012456.4 expression. (D) The low expression of lncRNA AC012456.4 was significantly associated with poor prognosis in patients with OSCC (p < 0.0001).

Figure 5 KEGG pathway enrichment analysis of lncRNA AC012456.4.

(A) Enrichment of genes in the KEGG MAPK SIGNALING PATHWAY by GSEA. (B) Heat map of core enrichment genes in the gene set KEGG MAPK SIGNALING PATHWAY. (C) Enrichment of genes in KEGG PATHWAYS IN CANCER by GSEA. (D) Heat map of core enrichment genes from the gene set KEGG PATHWAYS IN CANCER. (E) Enrichment of genes in KEGG OXIDATIVE PHOSPHORYLATION by GSEA. (F) Heat map of core enrichment genes from the gene set KEGG OXIDATIVE PHOSPHORYLATION. (G) Enrichment of genes in KEGG SPLICEOSOME by GSEA. (H) Heat map of core enrichment genes from the gene set KEGG SPLICEOSOME. The GSEA software was used to calculate enrichment levels.

Figure 6 GSEA were carried out to identify upregulated or downregulated GO.

(A) Enrichment of genes in GO ADAPTIVE IMMUNE RESPONSE by GSEA. (B) Heat map of core enrichment genes in the gene set GO ADAPTIVE IMMUNE RESPONSE. (C) Enrichment of genes in GO POSITIVE REGULATION OF CELL ACTIVATION by GSEA. (D) Heat map of core enrichment genes in the gene set GO POSITIVE REGULATION OF CELL ACTIVATION. (E) Enrichment of genes in GO RRNA METABOLIC PROCESS by GSEA. (F) Heat map of core enrichment genes in the gene set GO RRNA METABOLIC PROCESS. (G) Enrichment of genes in GO RIBOSOME BIOGENESIS by GSEA. (H) Heat map of core enrichment genes in the gene set GO RIBOSOME BIOGENESIS. The GSEA software was used to calculate the enrichment levels.

Relationship between lncRNA AC012456.4 and biological pathways and functions

Biological pathways and functions of lncRNA AC012456.4 were identified by GSEA. This analysis revealed that lncRNA AC012456.4 was involved in many critical pathways and correlated with tumorigenesis. A total of 150 pathways listed in the high-risk group were enriched, including KEGG MAPK SIGNALING PATHWAY, KEGG JAK-STAT SIGNALING PATHWAY, KEGG CALCIUM SIGNALING PATHWAY and KEGG PATHWAYS IN CANCER. Twenty-seven pathways in the low-risk group were also identified, including the KEGG OXIDATIVE PHOSPHORYLATION, KEGG PROTEASOME and KEGG SPLICEOSOME (Fig. 5). Similarly, 3073 GO annotations in the high-risk group and 516 GO annotations in the low-risk group were enriched (Fig. 6). Relevant partial results for KEGG pathways and GO analysis are listed in Table 6 and Table 7.

Table 6 KEGG Pathways enriched in high-risk and low-risk groups by using GSEA.

NAME	SIZE	ES	NES	NOM p-val	FDR q-val	FWER p-val	Rank at max	Leading edge	
KEGG_PRIMARY_IMMUNODEFICIENCY	35	0.783950	2.003367	0.002036	0.080199	0.032	5022	tags=63%, list=9%, signal=69%	
KEGG_CYTOKINE_CYTOKINE_RECEPTOR_INTERACTION	258	0.503302	1.751613	0.016227	0.351688	0.258	13393	tags=46%, list=23%, signal=60%	
KEGG_JAK_STAT_SIGNALING_PATHWAY	151	0.462485	1.585162	0.051020	0.356088	0.496	11252	tags=35%, list=19%, signal=43%	
KEGG_PATHWAYS_IN_CANCER	324	0.296756	1.015304	0.442386	0.570524	0.968	12772	tags=28%, list=22%, signal=36%	
KEGG_MAPK_SIGNALING_PATHWAY	265	0.353983	1.239951	0.226804	0.527153	0.881	11268	tags=29%, list=19%, signal=35%	
KEGG_PROTEASOME	46	−0.542264	−1.310828	0.249049	1	0.849	11204	tags=54%, list=19%, signal=67%	
KEGG_CYTOSOLIC_DNA_SENSING_PATHWAY	55	−0.342477	−1.059409	0.361581	1	0.958	6866	tags=33%, list=12%, signal=37%	
KEGG_SNARE_INTERACTIONS_IN_VESICULAR_TRANSPORT	38	−0.365953	−0.983674	0.481132	1	0.969	6863	tags=32%, list=12%, signal=36%	
KEGG_OXIDATIVE_PHOSPHORYLATION	118	−0.269338	−0.724989	0.681050	1	0.992	11643	tags=38%, list=20%, signal=48%	
KEGG_SPLICEOSOME	123	−0.362891	−0.936620	0.566473	1	0.978	5025	tags=24%, list=9%, signal=26%	

Table 7 GO annotation enriched in high-risk and low-risk groups by using GSEA.

Name	Size	ES	NES	NOM p-val	FDR q-val	FWER p-val	Rank at max	Leading edge	
GO_B_CELL_RECEPTOR_SIGNALING_PATHWAY	54	0.749803	1.963207	0.003838	0.974954	0.161	6389	tags=67%, list=11%, signal=75%	
GO_ADAPTIVE_IMMUNE_RESPONSE	279	0.614785	1.932954	0.007648	0.761521	0.202	7793	tags=46%, list=13%, signal=53%	
GO_NEGATIVE_REGULATION_OF_INTERLEUKIN _6_PRODUCTION	33	0.711452	1.897834	0	0.660863	0.28	10264	tags=67%, list=18%, signal=81%	
GO_REGULATION_OF_B_CELL_ACTIVATION	121	0.626420	1.886616	0.003883	0.617897	0.294	9579	tags=55%, list=16%, signal=65%	
GO_POSITIVE_REGULATION_OF_CELL_ACTIVATION	305	0.540650	1.725436	0.031496	0.464684	0.631	11768	tags=47%, list=20%, signal=58%	
GO_CELLULAR_RESPONSE_TO_ZINC_ION	16	−0.60868	−1.550511	0.056310	1	0.883	4440	tags=56%, list=8%, signal=61%	
GO_RIBOSOMAL_LARGE_SUBUNIT_BIOGENESIS	48	−0.60318	−1.496404	0.109343	1	0.925	5330	tags=42%, list=9%, signal=46%	
GO_POSITIVE_REGULATION_OF_PEPTIDYL_SERINE _PHOSPHORYLATION_OF_STAT_PROTEIN	21	−0.52630	−1.392874	0.115079	1	0.962	6411	tags=48%, list=11%, signal=53%	
GO_RRNA_METABOLIC_PROCESS	249	−0.38387	−1.055244	0.457925	1	0.998	10606	tags=36%, list=18%, signal=44%	
GO_RIBOSOME_BIOGENESIS	300	−0.38284	−1.050548	0.456692	1	0.998	11706	tags=38%, list=20%, signal=47%	

Discussion

OSCC is a common, highly invasive type of oral cancer prone to early recurrence and metastasis (Massano et al., 2006; Singh & Schenberg, 2013). Therefore, early diagnosis and treatment of OSCC is essential (Bozec et al., 2009). While cytology- and pathology-based methods have been applied to the clinical differential diagnosis of OSCC, limitations in the detection methods and poor prognoses have limited the five-year survival rate (Omar, 2013). Hence, more reliable, accurate and sensitive prognosis biomarkers and tools for early diagnosis are urgently needed (Mehrotra & Gupta, 2011). In recent years, many studies have revealed a close association between aberrant expression of lncRNAs and tumorigenesis (Alessandro & Irene, 2014; Batista & Chang, 2013; Espinosa, 2017; Rinn & Chang, 2012; Slaby, Laga & Sedlacek, 2017), which may aid in cancer diagnosis and prognosis.

Fewer than 2% of genes in the human genome are transcribed, and up to 98% of these transcripts are non-coding RNAs (Jandura & Krause, 2017; Espinosa, 2013; Quinn & Chang, 2016). lncRNAs are a class of non-coding transcripts ≥ 200 nucleotides in length that are actively involved in many biological processes, such as epigenetic regulation, cell cycle regulation, chromatin modulation and regulation of multiple gene expression (Rinn & Chang, 2012; Wang et al., 2017). These non-coding transcripts also play key roles in the occurrence, development and progression of malignant tumors (Espinosa, 2017; Kopp & Mendell, 2018; Spizzo et al., 2012). An increasing number of studies have reported that lncRNAs can play essential roles as oncogenes or tumor suppressor genes involved in the development and progression of various cancers (Batista & Chang, 2013; Espinosa, 2017; Kopp & Mendell, 2018; Reik, 2009; Rinn & Chang, 2012; Slaby, Laga & Sedlacek, 2017; Spizzo et al., 2012), including OSCC (Fang et al., 2017; Gomes et al., 2017; Guo et al., 2017; (Li et al., 2017). For example, the down-regulation of HOTAIR is associated with cancer progression in 26 human tumor types (Bhan & Mandal, 2015).

However, most early studies focused on a single gene or the results obtained from a single cohort study of lncRNAs and OSCC. Sun et al. (2017) used qRT-PCR to analyze the expression levels of lncRNA PDIA3P in 58 OSCC and paired noncancerous tissue samples. This study found that the overexpression of lncRNA PDIA3P correlated with lower survival rates for OSCC patients. One study by Wu et al. (2015) suggested that high expression of lncRNA HOTAIR in OSCC patients would contribute to the development and progression of cancer, leading to a poor prognosis. Similarly, LINC00668 expression is increased in both 50 OSCC tissues and cells, and over-expression is significantly correlated with poorer survival for OSCC patients; Therefore, this might be a negative predictive factor for the prognosis of OSCC patients (Zhang, 2017). In the era of big data, the development of TCGA and GEO technology has allowed researchers to predict and identify new biomarkers, which has enhanced the reliability and accuracy of current research. Cui et al. (2017) used TCGA and GEO data to determine that the expression levels of several lncRNAs, including RP1-228H13.5, TMCC1-AS1, LINC00205, and RP11-307C12.11, were associated with OS and recurrence-free survival of hepatocellular carcinoma patients. Three lncRNAs (LINC01140, TGFB2-OT1, and RP11-347C12.10) were significantly correlated with prognoses of hepatocellular carcinoma patients, independent of some clinical characteristics. Using the database, three lncRNAs, which may play key roles in the development, progression, and recurrence in gastric cancer, were identified (Song et al., 2017). However, the functions, roles, and molecular mechanisms of lncRNAs associated with OSCC remain unclear.

In this study, we identified lncRNAs that are dysregulated in OSCC and evaluated the relationships between the TCGA database and the clinicopathological features of these OSCC patients. Based on the above analysis, a total of 21 lncRNAs were correlated with patient prognoses, of which 13 lncRNAs (TTC39A-AS1, RP11-93B14.9, AC012456.4, RP11-87C12.5, RP11-464F9.21, LINC01549, RP11-897M7.1, AP003900.6, LINC01343, RP11-181E10.3, CTD-2545H1.2, RP11-796E2.4 and LINC01108) were significantly positively associated with OS and DFS, while the up-regulation of the latter eight lncRNAs (AC007879.2, BOK-AS1, CTB-161M19.4, CTD-2033A16.3, FAM95B1, RP11-1C8.7, RP11-285G1.14 and RP11-286E11.1) were correlated with poorer prognoses. Lan et al. (2017) have also reported that RP11-1C8.7 predicted the progression and outcome of patients with kidney renal papillary cell carcinoma and was regarded as an independent prognostication factor for kidney renal papillary cell carcinoma. Thus far in the published literature, no report has evaluated the biological function and molecular mechanisms of other lncRNAs associated with human cancers.

To our knowledge, this study is pioneering research and identified the lncRNA AC012456.4, which exhibited significantly lower expression in OSCC tissues than in adjacent normal tissues. Additionally, a Kaplan–Meier survival analysis (Gyorffy, Lánczky & Szállási, 2012) as well as univariate and multivariate Cox regression analyses revealed that lncRNA AC012456.4 was an independent prognostic factor and was significantly correlated with shorter OS and DFS. Further validation via the GEO database was consistent with the TCGA database analysis results. Moreover, we further evaluated the relationship between AC012456.4 expression and the clinicopathological features of OSCC patients. Low levels of AC012456.4 were found to be significantly associated with the history of alcohol consumption in OSCC patients. Interestingly, according to previous studies, we found that alcohol consumption can increase the probability of G:C to A:T transitions and that alcohol drinkers exhibited a significantly higher incidence of p53 mutations in OSCC (Hsieh et al., 2001), which suggested that alcohol may play a critical role in the progression of OSCC.

Since lncRNAs perform their biological function by specifically binding to target genes, we further explored the possible biological functions and molecular pathways of AC012456.4. Through GSEA, AC012456.4 was found to be significantly involved with tumor-related signaling pathways and crucial biological functions in tumorigenesis. Key pathways and functions for tumor initiation and progression were identified, such as GO biological function annotation and KEGG pathways, including the adaptive immune response, RRNA metabolic processes, CALCIUM, MAPK, and the JAK/STAT signaling pathway. Additionally, mutation, aberrant expression and modification of these GO annotations and signaling pathways have been frequently reported in OSCC and other cancers. We found that the MAPK pathway could be activated by the low expression of the tumor suppressor QKI-5, which can promote the proliferation of OSCC cells (Fu & Feng, 2015). We also revealed the strong relationships between HOXC10 and gastric cancer cell proliferation and metastasis, which occur through the MAPK pathway (Guo et al., 2017). Other pathways and biological functions have also been reported in pancreatic ductal adenocarcinoma (Huang et al., 2017a), hepatocellular carcinoma (Huang et al., 2017b; Wonganan et al., 2017), and human papillomavirus-transformed tumors (Skeate et al., 2018).

Dysregulated expression of lncRNA signatures has tremendous potential value, but this research has limitations. Above all, we have explored the correlation between AC012456.4 expression and OSCC prognosis based on the TCGA and GEO databases, which signifies that the exploration was performed using a bioinformatics approach. Then, further research, such as quantitative real-time PCR, as well as in vivo and in vitro experiments, will require collaborative efforts to explore the potential molecular functions and related mechanisms of these lncRNAs in OSCC.

Conclusions

In summary, this study was the first to discover that lncRNA AC012456.4 was poorly expressed in OSCC, with decreased survival rates for OSCC patients. This may be a potential novel, independent biomarker and therapeutic target for the early diagnosis, pathological classification, clinical treatment and outcome prediction for OSCC. Nevertheless, these assumptions require validation and confirmation by larger, multicenter studies.

Supplemental Information

Table S1 All patients characteristics in OSCC from the TCGA database

The clinicopathological features of all OSCC patients from the TCGA database.

Click here for additional data file.

Table S2 Univariate and multivariate Cox regression analysis in patients with OSCC

Univariate and multivariate Cox regression analyses were applied to evaluate the 21 candidate lncRNA signatures as independent prognostic variables.

Click here for additional data file.

Table S3 Expression of AC012456.4 in normal tissues and OSCC tissues

AC012456.4 expression is significantly down-regulated in OSCC and normal tissue samples in the TCGA dataset.

Click here for additional data file.

Table S4 AC012456.4 expression and clinicopathological characteristics of patients with OSCC from the TCGA database

The relationship between the expression of lncRNA AC012456.4 OSCC tissues with clinicopathological parameters.

Click here for additional data file.

Dataset S1 Kaplan–Meier survival analyses and log-rank tests for DFS in OSCC

Kaplan–Meier survival analysis with the log-rank test was used to identify relationships between the above 2493 lncRNA signatures and OSCC patient survival. Then, we determined the levels of 126 lncRNA signatures that were significantly related to DFS.

Click here for additional data file.

Dataset S2 Kaplan–Meier survival analyses and log-rank tests for OS in OSCC

Kaplan–Meier survival analysis with the log-rank was used to identify relationships between the above 2493 lncRNA signatures and OSCC patient survival. Then, we determined the levels of 151 lncRNA signatures that were significantly related to OS.

Click here for additional data file.

Dataset S3 21 lncRNA signatures that were significantly related to OS and DFS

Kaplan–Meierlan-Meier survival analysis with the log-rank was used to identify relationships between the above 2493 lncRNA signatures and OSCC patient survival. Then, we determined the levels of 21 lncRNA signatures that were significantly related to OS and DFS.

Click here for additional data file.

Dataset S4 KEGG Pathway enrichment analysis by GSEA

Biological pathways and functions of lncRNA AC012456.4 were identified by GSEA. This analysis revealed that lncRNA AC012456.4 was involved in many critical pathways and correlated with tumorigenesis.

Click here for additional data file.

We thank the patients and investigators who participated in TCGA Research Network (http://cancergenome.nih.gov/), which provides a Web resource for exploring, visualizing, and analyzing multidimensional cancer genomics data. In addition, Xuegang Hu wants to thank, in particular, the patience, care and support from Yao Xiong, Qun Li, Jing Lin, Qin Hu, Tingting You, and Qian Xiong in lab 302 over the past years. The authors also acknowledge the significant advice from Dr. Shan Jiang.

Abbreviations

LncRNAs long non-coding RNAs

OSCC Oral squamous cell carcinoma

HR hazard ratio

CI confidence interval

DFS disease-free survival

OS overall survival

TCGA The Cancer Genome Atlas

GEO Gene Expression Omnibus

GSEA Gene Set Enrichment Analysis

KEGG the Kyoto Encyclopedia of Genes and Genomes

GO Gene Ontology

STDEV standard deviation

Additional Information and Declarations

Competing Interests

Author Contributions

Data Availability

The authors declare there are no competing interests.

Xuegang Hu conceived and designed the experiments, performed the experiments, analyzed the data, contributed reagents/materials/analysis tools, prepared figures and/or tables, authored or reviewed drafts of the paper, approved the final draft.

Zailing Qiu conceived and designed the experiments, performed the experiments, analyzed the data, contributed reagents/materials/analysis tools, prepared figures and/or tables, approved the final draft.

Jianchai Zeng performed the experiments, analyzed the data, prepared figures and/or tables, approved the final draft.

Tingting Xiao performed the experiments, analyzed the data, contributed reagents/materials/analysis tools, prepared figures and/or tables, approved the final draft.

Zhihong Ke performed the experiments, contributed reagents/materials/analysis tools, prepared figures and/or tables, approved the final draft.

Hongbing Lyu conceived and designed the experiments, authored or reviewed drafts of the paper, approved the final draft.

The following information was supplied regarding data availability:

The raw data are provided in a Supplemental File.

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
