# Peer review of "A novel long non-coding RNA, AC012456.4, as a valuable and independent prognostic biomarker of survival in oral squamous cell carcinoma"

_PeerJ, doi:10.7717/peerj.5307_

## Round 0.1 · original submission · Major Revisions

Please carefully address all critical points of both reviewers and revise your manuscript accordingly. You also should improve the English and grammar as recommended by the reviewers.

Reviewer 1 ·

Basic reporting

Some minor grammar problems.

Experimental design

“A novel long non-coding RNA, AC012456.4, as a valuable and independent prognostic biomarker of survival in oral squamous cell carcinoma” by Hu et al. is a paper that takes on the task of using the TCGA data on oral cancer and using that do determine lincRNAs associated with patient survival post diagnosis. This is a great idea to use the TCGA data set of OSCC to look for a link between lncRNAs and patient outcomes.
One question is why the authors use Significance Analysis of Microarrays to determine differentially expressed RNAs? This method has largely been replaced by DeSeq2 and a number of other approaches that do not produce the high number of false positives seen with SAM. The differential expression needs to be done with another method besides SAM, a method which is more reliable.

Line 105 false discovery rate was controlled at approximately 1% - Please eliminate approximately and should probably put 0.01 not 1%.
Figure 3 What is the y axis- log base 2 or 10? It should be labeled.

There needs to be more explanation of the Gene Set Enrichment Analysis. My understanding is that subjects with short survival were grouped together to find lncRNAs that correlated with that. Out of that lncRNA AC012456.4 was associated with poor survival. If GSEA is then run on the mRNAs that were enriched in the lncRNA AC012456.4 enriched samples in the same dataset then it is really just a search of molecular pathways associated with short survival. To say that it is associated with lncRNA AC012456.4 is misleading and requires a different set of samples, a validation set, to do the GSEA, if I understand how this was done.
Line 270 We found that AC012456.4 expression was significantly down-regulated in OSCC tissue when compared with matched adjacent non-cancerous tissues… This is redundant as it was just said in line 269. And since all the RNAs tested for survival correlation were already shown by SAM
to be differntailly expressed in the dataset it is to some degree circular reasoning. Though it is gratifying that it decreases with OSCC.
This paper would be much better if 100 samples could be set aside to use as a validation set for the survival analysis. Otherwise it is quite speculative. If that is not possible and the paper is accepted I see no value in the GSEA done the way it was done unless a second dataset is used or another approach is used.

Table 3 Univariate and Multivariate Cox regression analysis …
What is the rational of doing age specific analysis of Stage and tumor size? I think there may need to be more validation of why this was done.

Validity of the findings

Due to lack of validation set some of the findings are speculative.

Reviewer 2 ·

Basic reporting

Method is poorly written and should be re-rewritten and has to include the algorithm used for differential expression analysis.
the followings are examples method that should be revised:
“Download the R and SAM software paceages from the Stanford University website, then install and import into Microsoft Excel. Then, unzip the original data paceage into TXT text file format. Open this in Microsoft Excel and sort the data into the format required for analysis as instructed for the SAM software.”
“lncRNA data for tumor and normal tissue samples with fluorescence signal values of 0 must first be removed.”

What did the authors mean by the following sentence? What is the exact level for FDR? What they mean by saying “the change in fold was controlled more than twice?”
“The false discovery rate (FDR) was controlled at approximately 1%, and the change in fold value was controlled more than twice.”

The majority of discussion is the repetition of the results and does not provide any perspective on the methodology and findings and significance of the findings in relation with other studies.

Experimental design

The authors should provide the information about the definition of each variable. For example, how alcohol habit has been defined?
Smoking is correlated with Head and neck cancer. Authors should justify why they did not include smoking in their univariate and multivariable analysis.
How the author defined the cut-off for expression of lncRNA at 2? Did the author perform sensitivity analysis to check for the other cut-off and possible misclassification?
Table 3: Which variables were included in the multivariable analysis. Those variables and their definition should be mentioned under the table.
Table 5: What was the cut-off value for the category of “decreased number” and “non-decreased number”.

Validity of the findings

NA

Additional comments

English and grammar should be improved

---

## Round 0.2 · Minor Revisions

Although your revised manuscript was significantly improved, as per reviewer's comments there are several minor issues that remain to be fixed. Please address critiques and revise your manuscript accordingly.

Reviewer 2 ·

Basic reporting

The authors addressed some of the concerns regarding the manuscript, however, several issues are still remains. Authors stated in the revision : "The false discovery rate (FDR) was controlled at 0.01 (the fold change>2)." FDR is completely independent of fold change. what the author means by this sentence. The authors defined the stagings in the cover letter, however, they should also include that under the tables that classify patients based on staging. The authors stated that they defined the cut-off expression of lnc-RNA at 2 based on the publication of Sun et al (PMID: 27651312), however, they should find the cut-off based on sensitivity/specificity analysis derived from their own data to classify the patients.

Experimental design

NA

Validity of the findings

The authors stated that they defined the cut-off expression of lnc-RNA at 2 based on the publication of Sun et al (PMID: 27651312), however, they should find the cut-off based on sensitivity/specificity analysis derived from their own data to classify the patients.

---

## Round 0.3 · accepted · Accept

Thank you for addressing final critical points. Your manuscript is adequately revised and is recommended for publication now.

#